# Transient Receptor Potential (TRP) Channels in Airway Toxicity and Disease: An Update

**DOI:** 10.3390/cells11182907

**Published:** 2022-09-17

**Authors:** Isabel Müller, Philipp Alt, Suhasini Rajan, Lena Schaller, Fabienne Geiger, Alexander Dietrich

**Affiliations:** Walther-Straub-Institute of Pharmacology and Toxicology, Member of the German Center for Lung Research (DZL), LMU-Munich, Nussbaumstr. 26, 80336 Munich, Germany

**Keywords:** lung, alveoli, bronchi, pulmonary vasculature, TRPA1, TRPC6, TRPM2, TRPM5, TRPV2, TRPV4

## Abstract

Our respiratory system is exposed to toxicants and pathogens from both sides: the airways and the vasculature. While tracheal, bronchial and alveolar epithelial cells form a natural barrier in the airways, endothelial cells protect the lung from perfused toxic compounds, particulate matter and invading microorganism in the vascular system. Damages induce inflammation by our immune response and wound healing by (myo)fibroblast proliferation. Members of the transient receptor potential (TRP) superfamily of ion channel are expressed in many cells of the respiratory tract and serve multiple functions in physiology and pathophysiology. TRP expression patterns in non-neuronal cells with a focus on TRPA1, TRPC6, TRPM2, TRPM5, TRPM7, TRPV2, TRPV4 and TRPV6 channels are presented, and their roles in barrier function, immune regulation and phagocytosis are summarized. Moreover, TRP channels as future pharmacological targets in chronic obstructive pulmonary disease (COPD), asthma, cystic and pulmonary fibrosis as well as lung edema are discussed.

## 1. Introduction

The respiratory system of our body is not only essential for oxygen supply and removal of CO_2_, but is also exposed to toxicants and pathogens from both sides: the airways and the vasculature. Bacteria, viruses, chemical compounds, particulate matter (PM) and radiation can seriously harm respiratory function. Consequently, immune cells invade tissue as the first line of defense, but are also responsible for tissue damage by inflammation, resulting, amongst other diseases, in the acute respiratory distress syndrome (ARDS) [1]. Wound healing, involving fibroblast-to-myofibroblast differentiation and its proliferation, may turn pathological, inducing lung fibrosis with a life-threatening reduction in gas exchange [2]. 

Transient receptor potential (TRP) proteins as functional channels are important for cellular ion (mainly Ca^2+^ and Na^+^) homeostasis [3,4], which is also essential for multiple functions in different cells of the respiratory tract [5]. They may also be involved in the detection of and defense against hazardous compounds in the inhaled air or the perfused blood [6]. At present, 28 different mammalian TRP channels have been identified, comprising 6 TRP families (TRPA (for ankyrin with one isoform), TRPC (for classical with 7 isoforms), TRPM (for melastatin with 8 isoforms), TRPML (for mucolipidin with three isoforms), TRPV (for vanilloid with 6 isoforms), TRPP (for polycystin with three isoforms). TRP proteins are composed of intracellular N- and C-termini, 6 membrane-spanning helices (S1–S6), and a presumed pore-forming loop (P) between S5 and S6 (reviewed in [3]). Most of the TRP family members harbor an invariant sequence, the TRP box (containing the amino acid sequence: EWKFAR), in their C-terminal tails as well as ankyrin repeats in their N-termini [3]). For a functional TRP ion channel complex, four monomers of the same type in a homotetrameric complex or four different TRP monomers forming a heterotetrameric channel are essential (reviewed in [4]). This review will summarize TRP functions in the airways, their roles in diseases, such as chronic obstructive pulmonary disease (COPD), asthma, cystic and pulmonary fibrosis as well as lung edema. Our manuscript also serves as an update of recent reviews by us [5,6,7,8] and will focus on non-neuronal tissues and extensively studied TRP members, such as TRPA1, TRPC6, TRPM2, TRPM5, TRPV2 and TRPV4.

## 2. Cells and Their TRP Expression in the Trachea and Bronchi

The air, entering the mouth and trachea, is distributed first through a highly branched tree-like system called bronchi, which is lined by the so-called pseudostratified epithelium and supported by cartilage [9]. Subsequently, it enters the smaller branches lacking cartilage, which are known as bronchiole [9]. The pseudostratified epithelium is composed of self-renewable basal cells [10], which differentiate, among others, to club cells [11] with the capability to form goblet [12], brush [13] and ciliated cells [9,14] (see Figure 1). Both club and goblet cells produce mucins [15], which, together with the ciliary movements of ciliated cells, are essential for mucociliary clearance as the primary defense mechanism of the respiratory tract [16]. As bronchi become smaller, they lose cell diversity, with only club and ciliated cells lining the epithelium [9].

**TRPV4,** as the fourth member of the vanilloid family of TRP channels, is activated by both physical (osmolarity, mechanical stress and higher temperatures) and chemical stimuli, which may be endogenous, plant derived or synthetic ligands mainly developed by the GlaxoSmithKline (GSK) company (reviewed in [17,18,19]). In an elegant study, Valverde and his colleagues nicely showed the expression and function of TRPV4 in ciliated cells ([20] and see Table 1). They were able to link Ca^2+^ entry through TRPV4 channels with an ATP-induced increase in ciliary beat frequency (CBF). TRPV4 activator, like 4-phorbol 12,13-didecanoate (4PDD), showed no increase in intracellular Ca^2+^ and CBF in ciliated cells of TRPV4−/− mice in contrast to wild-type (WT) cells [20]. The autoregulation of CBF in response to high viscosity solutions, however, was preserved in TRPV4−/− cells despite a reduced Ca^2+^ signal [20]. Similar results were observed in human nasal epithelial cells with a more specific TRPV4 agonist, GSK1016790A, which, however, also induced the cessation of ciliary beating and cell death. This lethal effect was prevented by the TRPV4 antagonist HC067047 [21]. TRPV4 also mediates CBF by lipopolysaccharides (LPS), which is a major component of the membranes of Gram-negative bacteria [22]. If this protective response is missing in TRPV4−/− mice, they display exacerbated ventilatory changes and the recruitment of polymorphonuclear leukocytes into the airways [22].

Next to their functions described above, club cells also produce and secrete a surfactant composed of the two surfactant proteins, A and D (SP-A, SP-D) [23]. The lung surfactant was originally identified as complex, mainly consisting of lipids and a smaller amount (≈8%) of proteins facilitating alveolar gas exchange [24]. Especially SP-A and D were later also recognized as essential for host defense in the airways [23,25,26]. Single-cell RNA seq recently revealed the expression of TRPV4 channels in club cells [27] but their role for SP production, self-renewal and differentiation of the pseudostratified epithelium is still elusive. However, TRPV4 in club cell was recently linked to allergic airway inflammation ([28] and see Section 2.2).

**TRPM5** channels as Ca^2+^-activated members of the melastatin family of TRP proteins mediate signaling taste and other chemosensory cells (reviewed in [29]). In the respirato-ry tract, TRPM5 was identified in solitary chemosensory (SCC) [30] and microvillus cells [31] in the upper and in brush (or tuft) cells in the lower airways [32,33] (see Figure 1). Recent sequencing studies revealed that TRPM5 is a brush cell marker and a key signaling transduction molecule in SCC and airway brush cells [34]. A complete novel role for the TRPM5 signal transduction cascade in response to bitter tastants and to bacterial quorum-sensing molecules (QSM) [35] as well as to virulence-associated formyl peptides [36] was revealed in two current studies. Compounds, such as the bitter tasting denatonium acting via taste receptors (Tas2R) [37,38], *Pseudomonas aeruginosa* quorum-sensing molecules (QSM), such as Pseudomonas quinolone signal (PQS) and acyl homoserine lactones [35,39], as well as formylated bacterial peptides by unknown receptors [36], stimulated phospholipase C-β2 (PLCβ2) to produce inositol-tris-phosphate (IP3), which releases Ca^2+^ from internal endoplasmic reticulum stores. TRPM5 channels are directly activated by an elevation of intracellular Ca^2+^ ([Ca^2+^]_i_) ([40] reviewed in [29]), and the subsequent Na^+^ influx triggers release of the signaling molecule acetylcholine (ACh) from these cells. ACh acts as a paracrine on the muscarinic acetylcholine receptors 3 (M3R) in ciliated cells and enhances the ciliary beat frequency and the removal of particles from the airways [35,36,37]. Consequently, ciliated cells in TRPM5−/− mice showed reduced ciliary motility after the application of bacterial triggers [35,36]. Additionally, the TRPM5-mediated release of ACh from brush cells stimulates nicotinic ACh receptors on adjacent sensory nerve endings in the trachea to release the neuropeptides CGRP and substance P that mediates plasma extravasation, neutrophil recruitment and diapedesis [41]. Moreover, the QSM-mediated activation of TRPM5 triggers the secretion of immune mediators, among the most abundant members of the complement system, and impacts the early cytokine response in an onset of *P. aeruginosa* infection. The increase in IL-1, IL-6, KC, MCP-1, G-CSF and eotaxin was abolished in TRPM5−/− mice after infection. Consistently, infection with *P. aeruginosa* was more severe in TRPM5−/− mice, with more TRPM5−/− mice dying during the first 3 days of infection [39]. More recently, the transcriptome analyses of TRPM5-expressing microvillous cells indicated that they are likely involved in the inflammatory response elicited by viral infection of the olfactory epithelium [42]. Very intriguingly, TRPM5-expressing ectopic chemosensory cells were detected after influenza virus-induced injury [43] in the distal lung, though these cells are absent in healthy mice. The function of the channel in these cells is currently unknown. While human sinonasal epithelial cells in air–liquid interface cultures were shown to involve TRPM5 for the response to bitter tastants, the role of TRPM5 in human brush (tuft) cells needs to be assessed.

Trachea and bronchi also contain layers of smooth muscle cells (see Figure 1), which are important for bronchoconstriction in allergic airway responses (see Section 2.2) and express mainly members of the classical TRP (**TRPC**) family (TRPC1, 3 and 6, see [44]). All of the six channels, except TRPC1, are activated by diacylglycerol (DAG), which is produced by receptor-activated phospholipases-C (reviewed in [45,46]). Most interestingly, the deletion of TRPC6 in TRPC6−/− mice resulted in a compensatory up-regulation of TRPC3 channels in tracheal smooth muscle cells and a subsequent increase in allergic bronchoconstriction ([44] and see Section 2.2).

### 2.1. TRP Channels and Cystic Fibrosis (CF)

An aberrant viscous mucus, which is not able to remove particles and pathogens, is a major symptom in patients with CF (reviewed in [47]). CF patients carry mutations in the gene for the cystic fibrosis transmembrane conductance regulator (CFTR), an ion channel managing the passage of chloride and bicarbonate ions across the apical membrane of epithelial cells. Identified mutations in CF, including the most common F508del type, result in improper translation, processing and translocation of the CFTR protein to the plasma membrane as well as impaired conductance and regulation of the ion channel [48]. Due to the missing CFTR protein, the produced mucus lacks water and is populated mainly by *Pseudomonas aeruginosa*. Bronchial epithelial cells of CF patients, which are exposed to *P. aeruginosa*, express high levels of IL-8 often in co-expression with TRPA1 as the only members of the TRPA family. Notably, TRPA1 activation was shown to increase the release of IL-8 [49]. Silencing of TRPA1 and pharmacological inhibition of the channel in fibrotic primary bronchial epithelial cells after exposure to P. aeruginosa significantly reduced the expression of IL-8 as well as several other pro-inflammatory cytokines, such as TNF-alpha, IL-1beta and IL-6. These data suggest an important role for TRPA1 in mediating the immune reaction of CF patients and point to TRPA1 as a possible druggable target in CF [50]. The resulting lung inflammation can lead to serious lung damage and respiratory failure (reviewed in [47]). 

Several reports showed direct and indirect interactions of CFTR with TRP channels in the airway epithelium (summarized in [51]). CFTR down-regulates TRPC6-mediated Ca^2+^ influx, while TRPC6 up-regulates CFTR-mediated Cl- transport, and both proteins physically interact with each other [52]. Thus, TRPC6-mediated Ca^2+^ influx was increased in CF versus non-CF human epithelial cells because the functional coupling of CFTR and TRPC6 is lost. This mechanism called reciprocal coupling has also been observed in freshly isolated ciliated epithelial cells [52], which regulate mucus viscosity by CFTR activity. 

The seventh member of the melastatin family of TRP channels, TRPM7, is ubiquitously expressed and permeable not only for Ca^2+^, but also for Mg^2+^ and Zn^2+^ ions (summarized in [53]). Application of a specific TRPM7 channel activator, naltriben [54], resulted in a decreased CFTR function in HeLa cells heterologously expressing WT or F508del mutant CFTR channels. However, when another CFTR mutant (G551D) was expressed, an increased activity was detected [55]. The authors suggested naltriben as a new potentiator in patients with the G551D-CFTR mutation, although the relevance of their results in native airway epithelial still needs to be confirmed. 

Most interestingly, TRPV4 channels are activated by hypotonic solutions [17] and are responsible for the swelling-activated Ca^2+^ entry, which is essential for the regulatory volume decrease (RVD) in tracheal epithelial cells [56]. Along this line, Valverde and coworkers also demonstrated that the impaired RVD response in CF airway epithelia is caused by the misregulation of TRPV4, suggesting that the hypotonic activation of TRPV4 channels is CFTR dependent [56]. 

The sixth member of the vanilloid family of TRP channels, TRPV6, is highly selective for Ca^2+^ ions [57]. Inhibition of TRPV6 channels and an siRNA strategy suggested that TRPV6 was mostly involved in the increase in Ca^2+^ influx and upregulated in primary human airway epithelial cells from CF in comparison to non-CF cells [58].

Along with the detrimental production of non-functional mucus by epithelial cells, the immune function is seriously disabled in CF: macrophages (MΦ) and neutrophils are not able to defend against invading bacteria [59,60]. A drug called roscovitine, which acts as a partial corrector of the F508del CFTR protein [61] and recruits TRPC6 to phagosomal membranes of alveolar MΦ, is able to restore the microbicidal function compromised by CF [62]. However, a recent phase II study revealed no significant efficacy on inflammation, infection, spirometry, sweat chloride, pain and quality of life in roscovitine-treated groups compared to placebo-treated controls [63].

In conclusion, there is some evidence for the involvement of TRP channels in CF via interaction with CFTR, but the exact mechanisms are still not known. 

### 2.2. Non-Neuronal TRP Channels in Asthma and Airway Inflammation

Asthma is a chronic inflammatory disease of the airways induced by repeated exposure to specific allergens or other triggers (e.g., exercise and cold air), which results in the activation of epithelial cells and acute bronchoconstriction. Patients suffer from symptoms such as cough, dyspnea, wheezing, and chest tightness [64]. Among other cells of the immune system, mainly differentiated T-helper 2 (Th2) cells, mast cells, and eosinophils invade lung tissues. They are responsible for the up-regulation of mediators of allergic inflammation, such as immunoglobulin E (IgE), IL-4, IL-5, IL-13, eotaxin (CCL11), and eicosanoids, as well as increased mucus production (summarized in [64,65,66]). As a therapeutic option, asthma patients inhale rapid and long-acting β2-adrenoceptor agonists for bronchodilation and glucocorticoids to inhibit chronic inflammation [67]. These current treatment options offer only symptomatic relief to the majority, but not for all patients [68]. There are, however, no therapeutic options to date that are able to prevent or cure asthma [67]. 

Bronchoconstriction is a most prominent response to allergens during allergic asthma. Patients lose their capacity to exhale CO_2_ efficiently [65]. In a recent publication, TRPA1 channels were detected in human airway smooth muscle cells [69]. Importantly, GDC-0334 a highly potent, selective, and orally bioavailable TRPA1 antagonist-inhibited TRPA1 function in airway smooth muscle decreasing allergic airway inflammation [69]. This antagonist was also effective in a phase I study in humans against pain and itch most probably due to the blocking of TRPA1 activity in sensory neurons [69]. TRPC1, 3 and 6 mRNAs are also expressed in murine airway smooth muscle cells [44]. TRPC6-deficiency in mice, however, does not result in reduced allergic bronchoconstriction induced by methacholine or increased expiration rates after sensitization by ovalbumin (OVA) analyzed by head-out body plethysmography [44]. On the contrary, TRPC6−/− mice showed an increased airway constriction and reduced expiration rates compared to WT mice, which is most probably due to a compensatory up-regulation of TRPC3 channels [44]. IgE and typical Th2 cytokines, such as IL-5 and IL-13, however, were reduced, while mucus production was not changed in TRPC6−/− mice compared to WT controls [44]. Similar results were obtained for TRPC1-deficient mice challenged with OVA, which showed a significantly reduced allergen-induced pulmonary leukocyte infiltration coupled with an attenuated Th2-cell response [70]. As it is unclear if TRPC1 monomers are able to form a functional homo-tetrameric channel or rather work as channel regulators in hetero-tetrameric complexes (reviewed in [71]), these data favor a functional TRPC1/6 heteromeric channel [72] in cells involved in allergic airway inflammation. Moreover, a knock-down of TRPC3 protein by the intravenous injection of small hairpin RNAs specifically targeting TRPC3 in lentiviral particles or the application of channel blockers was also able to reduce allergy-induced airway disease [73], emphasizing the role for all three TRPC channels in allergic asthma. 

In a gene-expression study of childhood asthma induced by three allergens (spring pollen, dust mite, dog and cat hair) in North China, a 2.6-fold increase in TRPV2-mRNA in lymphocytes was revealed [74]. These results were confirmed in an OVA mouse model, where TRPA1 and TRPV1 gene expression was also up-regulated ([75] reviewed in [76]). The exact role of this channel, which is further described in Section 3.2, in this setting, however, remains elusive.

There are several lines of evidence for the important role of TRPV4 channels in asthma and inflammation of the airways. TRPV4-deficient mice were protected from airway remodeling in a house dust mite (HDM) model, which is more relevant to the human situation than OVA challenge [77]. Moreover, in allergic rhinitis caused by HDM, the up-regulation of TRPV4 proteins in nasal cells was demonstrated, which resulted in epithelial barrier disruption [78]. TRPV4 is also expressed in human airway smooth muscle cells (see Figure 1), and specific agonists induce the release of ATP in non-atopic, immunoglobulin E-independent asthma patients [79]. ATP is able to activate P2X4 receptors on mast cells releasing cysteinyl leukotrienes (cystLT1), which then causes the cystLT1-dependent contraction of airway smooth muscle cells [79]. Other allergens, such as the alkaline protease Alp1 from household molds, induce typical symptoms of asthma after inhalation [80]. In an elegant study, Klein and colleagues showed that Alp1 destroys the cell junctions of club cells. Consequently, a club-cell-specific TRPV4 deficiency resulted in the decreased production of the C-C motif chemokine ligand 2 (CCL2) and a reduction in immune cells after the inhalation of Alp1 in these mice in comparison to WT controls [28]. On a molecular level, the over-expression of TRPV4 channels in these cells resulted in a Ca^2+^/calcineurin-dependent increased Th2 response to Alp1 in the airways [28]. In a translational approach, the authors also reported that a single nucleotide polymorphism (SNP) rs6606743 in the human *Trpv4* gene increased the expression of the channel protein and is associated with fungal immunization and asthma in humans [28].

Importantly, a role for TRPV1, which is mainly expressed in neurons, for the secretion of IL-33 by the airway epithelium in response to HDM and fungal allergens was described recently [81]. Functions of this and other neuronal expressed TRP channels, such as TRPA1, TRPM3, and TRPM8, but also TRPV4 in the regulation of airway inflammation as polymodal sensors by the release of neuropeptides, which is not in the focus of this review, are nicely summarized in recent manuscripts [76,82,83]. 

### 2.3. Lung Toxicity and TRP Channels in the Trachea and Bronchi

Water-soluble, highly reactive compounds, such as acrolein and hydrochloric acid (HCl) as well as chlorine (Cl_2_), mainly act upon the upper part of the lower airways. Cl_2_ is used in disinfectants as well as a warfare agent and produces hypochlorous acid (HClO) after inhalation upon contact with water of the mucus in the respiratory tract. Hypochlorite-induced respiratory depression and pain behavior were reduced in mice lacking the TRPA1 channel in comparison to wild-type controls [84]. TRPA1 is predominantly expressed in neurons of the respiratory tract (reviewed in [85]), but may also be present-although in minor quantities-in non-neuronal tissues [86]. TRPA1 channels harbor up to 18 ankyrin repeats at its N-terminus and are activated via modification of their cysteine residues. For acrolein, as one of the compounds of inhaled cigarette smoke (>50 ppm), its electrophilic double bond reacts with the highly nucleophilic sulfhydryl group of a cysteine [87] in the N-terminus of TRPA1, triggering altered channel activity [88,89]. Exposure of mice to acrolein (100–275 ppm for 10–30 min) resulted, among others, in the sloughing of the airway epithelium and increased mortality [90]. Most interestingly, the treatment of mice with HCC030031, a TRPA1 inhibitor, resulted in a significantly reduced mortality ([90] reviewed in [85]). Moreover, other TRP channels, such as TRPA1 and TRPV1, were identified in a cigarette-smoke-induced airway epithelial cell injury model [91]. 

The accidental inhalation of HCl, acid regurgitation or formation of HCl by Cl_2_ and water of the mucus are the main triggers for acid-induced toxicity of the upper part of the lower respiratory tract. Although several TRP channels are acid sensitive (e.g., TRPC4, TRPC5, TRPV1, TRPV4 and TRPP2), only TRPV4 channels were extensively studied. In a mouse model of HCl-induced acute lung injury, two TRPV4 antagonists (GSK2220691 and GSK2337429A) were able to reduce the symptoms [92]. Importantly, also post-exposure channel antagonism reduced the critical hallmarks of chemical lung injury, such as airway hyperreactivity and increased lung elastance, protein leakage, and neutrophil and macrophage infiltration of the lungs ([92] reviewed in [85]). These results were confirmed by another research group. In this report, the prophylactic inhibition of TRPV4 with GSK2193874 or genetic deletion of the gene in TRPV4−/− mice attenuated HCl-induced lung injury, while post-exposure treatment with another antagonist (HC-067047) showed no benefit ([93] reviewed in [85]). In a mouse model mimicking the real chlorine exposure, the same TRPV4 antagonists, which suppressed injuries by HCl, also reduced Cl_2_-induced damages [92]. Moreover, neutrophils expressing TRPV4 channels seem to be involved in the inflammatory response to Cl_2_ (reviewed in [85]). 

Inhaled particulate matter (PM) can induce acute and chronic toxicity in the respiratory system. However, target sites differ depending on particle size and water solubility. The respiratory system is mainly affected by fine PM of an average particle size <5 µm in diameter (reviewed in [5,94]). 

PM is produced by biomass and fossil fuel burning, but also by friction from tires and brakes and natural sources, such as sandstorms and wildfires. Moreover, it is artificially generated, e.g., in biotechnology, food industry, cosmetics and electronics (reviewed in [95]). Larger particles trapped in the mucus coating the epithelium of the upper part of the lower airways can still be removed by mucociliary clearance (see above) and initiation of the cough reflex [5]. As outlined above the supposedly mechanosensitive TRPV4 channel is expressed in ciliated cells regulating CBF and is therefore a good candidate for protecting the airways from inflammation induced by larger particles. However a direct evidence for its mechanical activation by PM > 5 µm is still missing [95].

Ultrafine particulate matter is determined to have an average diameter of <0.1 µm. Ultrafine particles (UFP) are able to penetrate cellular structures, organelles and enter the vascular system [84]. The nanoparticles (NP) silicium dioxide (SiO_2_) and titanium dioxide (TiO_2_) are of particular interest, as they make up a significant fraction of industrial PM. The interaction of these inorganic nanoparticles with TRP channels, e.g., TRPV1 and TRPA1 (reviewed in [96]), have been addressed in several studies. Sanchez et al. showed that colloidal SiO_2_ (silica) nanoparticles (SiNP) increase the response of TRPV1 to its agonist, capsaicin, in HEK293T cells. In addition, SiNP inhibited TRPV4 in human bronchial epithelial cells in a concentration-dependent manner. This was confirmed both in murine tracheal epithelial cells and murine TRPV4-overexpressing HEK293T cells. Furthermore, a moderate decrease in the ciliary beat frequency after SiNP application was found in murine tracheal epithelial cells [97]. However, in another study, human TRPV4-overexpressing HEK293 cells showed no altered cytotoxicity after the application of various nonporous silica nanoparticles [98].

Titanium dioxide (TiO_2_) nanoparticles (TiNP) are known to be less potent than silica nanoparticles [99]. Nevertheless, they induced emphysema-like lung injury in mice with significant changes in morphology and histology only one week after the intratracheal instillation of 0.1 mg TiNP [100]. Another research group showed in a mouse model for asthma that, among others, TRPV4 protein levels are increased after nanoTiO_2_ inhalation [101]. Furthermore, TiO_2_ application to human bronchial epithelial cells increased mucin secretion via a Ca^2+^ signaling pathway [100]. The contrast in the effects of SiNP and TiNP on TRP channels is indicative of the broad interaction profile of ultrafine particulate matter.

Diesel exhaust particles (DEP) are a mixture of numerous nanoparticles, heavy metals and chemicals attached to a carbonaceous core. The variety of hazardous chemicals causes severe and pleiotropic damage to human airways. Among others, one reactive species in DEP is acrolein. In addition, Li et al. showed that DEP exposure results in a TRPV4-mediated Ca^2+^ influx that finally leads to an increase in MMP-1 activation. They also discovered a TRPV4 signaling complex located in the cilia of human bronchial epithelial cells. This complex is activated by the chemical extract of DEP, and not by carbon, which suggests that the carbonaceous core of DEP might serve as a vehicle for the active species. The researchers also found that the genetic polymorphism TRPV4_P19S_ increases MMP-1 activation in response to DEP [102]. This polymorphism is also of clinical interest, as it confers a predisposition to the development of chronic obstructive pulmonary disease (COPD) [103]. TRPV4 channels appear to play essential roles in lung function and toxicant sensing, making them major therapeutic targets in the treatment of toxic injury of the respiratory tract.

The inhalation of toxicants and pollutants, such as ozone (O_3_), sulfur dioxide (SO_2_) and nitrogen oxides (NO_x_), can also contribute to particulate- and chemical-induced carcinogenesis (reviewed in [104]). Based on the initial defense reaction of inhalant-induced lung injury, reactive oxygen- and reactive nitrogen species (ROS/RNS), such as superoxide (•O_2_^−^), nitric oxide (•NO) or peroxynitride (•ONOO^−^), are formed. The production of pro-inflammatory cytokines and chemokines is triggered and initiates the recruitment of immune cells [104,105]. The chronic inhalation of toxicants and pollutants leads to an abnormal production of ROS/RNS that are known to interact with DNA in the proliferating epithelium and increases the risk of genomic alteration and DNA damage. The extensive DNA damage creates a chronically inflamed tissue that is rich in inflammatory cells, ROS/RNS, DNA damage, cell-proliferating growth factors and other growth supporting stimuli [106]. Chronic inflammation, including all the associated characteristics, is a hallmark of cancer development.

TRP channels, such as TRPV1, are activated by ROS induced by blue light [107] and can be blocked by its antagonist after UV exposure [108]. They also have been associated with growth stimuli. Epidermal growth factor (EGF) was found to be responsible for an increased surface expression of TRPM7 and an enhanced cell migration of A549 cells [109]. The silencing or chemical inhibition of TRPM7 led to a total inhibition of cell migration, which makes TRPM7 a potential target for the prevention of metastasis [109]. In addition, TRPC1 was identified in human non-small cell lung carcinoma (NSCLC) cells as a major regulator for EGF signaling that affects cell proliferation [110]. 

Aside from EGF signaling, TRP channels have been implicated in several aspects of cancer progression. In 2013, an involvement of TRPC1, 3, 4 and 6 in cell differentiation in NSCLC was discovered [111]. TRPC1 is highly expressed in NSCLC and facilitates tumor proliferation, cell migration and cell survival [112], and TRPV3 was reported to be overexpressed in NSCLC [113]. Along this line, the silencing of TRPV3 led to cell cycle arrest [113].

In addition to their role in cancer progression, TRP channels are known to promote resistance to chemotherapy in the treatment of cancer. TRPA1 is both overexpressed in small cell lung cancer (SCLC) and plays a role in cell survival since it is able to prevent apoptosis [114]. TRPV1 was not only upregulated in A549 cells, but after application of cisplatin and fluorouracil, also mediated resistance against these chemotherapeutics and promoted cell survival [115].

## 3. TRP Expression and Function in Alveoli

The bronchi and bronchioles terminate in millions of alveoli, which are very tiny, thin-walled and highly vascularized sacs facilitating gas exchange [9]. This section will focus on three alveolar cell types of major interest: alveolar type 1 (AT1), alveolar type 2 (AT2) cells and alveolar macrophages (AMΦ). AT1 cells are flat, cover most of the surface of alveoli and are highly permeable for water to ensure fluid homeostasis but are also essential for barrier function. Specific marker proteins of these cells are podoplanin as membrane glycoprotein and aquaporin-5, a water-conducting channel [116]. AT2 cells are a cubic and secret surfactant [117], which is necessary for a reduction in surface tension for easy gas exchange, as well as preventing a collapse of alveoli during breathing [118]. Moreover, AMΦ are able to phagocytize bacteria viruses and PM in alveoli which were not successfully removed in the upper part of the lower airways (see Figure 1).

### 3.1. Expression of TRP Channels in Alveolar Cells and Their Role in Barrier Function

We were able to show the expression of **TRPV4** in both AT1 and AT2 cells and identified emphysema-like changes in the lungs of older TRPV4−/− mice, which favor ischemia-reperfusion(I/R)-induced edema formation, a serious drawback in lung transplantation ([119] reviewed in [8]). AT2 cells are also able to replace AT1 cells, a capacity that was, however, not dependent on TRPV4 channels [119]. Most interestingly, the cell barrier function of AT2 cells differentiated to AT1 cells was significantly lower, which was identified by electrical cell-substrate impedance sensing (ECIS) [119]. Our results confirm earlier findings on the role of TRPV4 channels in barrier function of the skin [120], the urogenital tract [121] and the corneal epithelium [122]. The exact molecular role of TRPV4 channels in cell barrier function, however, still needs to be clarified.

### 3.2. TRP Channels and Alveolar Toxicity

Ultrafine PM with a size <0.1 µm mainly from diesel exhaust particles (DEP) and cigarette smoke may reach alveoli and can enter the vascular system [123]. Alveolar macrophages (AMΦ), which predominantly express **TRPV2** channels (see Figure 1), are able to eliminate these particles by phagocytosis. The second member of the vanilloid TRP family is a Ca^2+^ permeable channel activated by heat (>52 °C) and various ligands, including cannabinoids and mechanical stress (reviewed in [124,125]). TRPV2 channels are, in most cells, located in the endoplasmic reticulum, but upon stimulation with phosphoinositidylinositol 3-kinase-activating ligands, they are translocated to the plasma membrane (reviewed in [124]). Most interestingly, TRPV2-deficient mice exposed to cigarette smoke for two months showed alveolar space enlargement, which was absent in control mice [126]. These emphysema-like changes may be due to a defective phagocytosis in TRPV2−/− AMΦ compared to WT controls [126] and are another important feature of COPD in heavy smokers. Therefore, activation of TRPV2 channels may protect from cigarette smoke-induced COPD in alveoli. A similar mechanism was observed in TRPML3-deficient mice, which showed emphysema-like changes that were further exacerbated by exposure to tobacco smoke [127]. These changes were linked to an impaired early endolysosomal trafficking as well as defects in endocytosis in TRPML3−/− mice [127]. Our manuscript focuses on non-lysosomal TRP channels, but the role of lysosomal TRP channels in airway diseases was summarized in a recent comprehensive review [128].

As outlined above **TRPV4** channels are expressed in the alveolar epithelium [119], but their role in toxicant detection or in the resulting inflammation and lung injury is still elusive.

## 4. TRP Expression in Fibroblasts and Their Involvement in the Development of Lung Fibrosis

Pulmonary fibroblasts are most important in wound healing after acute and chronic lung injury. After lung injury, epithelial cells produce transforming growth factor β1 (TGF-β1) to initiate fibroblast-to-myofibroblast differentiation. Myofibroblasts express more α-smooth muscle actin (α-SMA) and secret an extracellular matrix, such as collagens, fibronectin and plasminogen activator inhibitor 1 (PAI-1). An impaired repair process after chronic lung damage by toxicants (e.g., the cytostatic drug bleomycin), however, is able to induce lung fibrosis, which is characterized by fibroproliferative foci, seriously inhibiting gas exchange. Pulmonary or lung fibrosis is a chronic progressive disease without effective medical treatment options, leading to respiratory failure and death within 3–5 years of diagnosis [2]. Two compounds are approved for the treatment of idiopathic lung fibrosis (IPF), pirfenidone, which down-regulates the production of growth factors such as TGF-β1 and procollagens and nintedanib, a multi-tyrosine kinase inhibitor. However, both therapies merely slow down the reduction in forced vital capacity and augment the patients’ quality of life, but there is no enhancement of survival rates [129,130]. 

In a fetal human lung fibroblast cell line (MRC5), down-regulation of the **TRPM7** protein was shown to decrease TGF-β1-induced collagen and α-SMA synthesis [131], but the role of the channel in native cells remains elusive.

Primary human and mouse fibroblasts also constitutively express **TRPV4** channels, which is not changed in patients suffering from IPF [132]. Importantly, TRPV4-deficient mice were protected from fibrosis and genetic ablation, or the pharmacological inhibition of TRPV4 function abrogated myofibroblast differentiation, which was restored by the heterologous expression of TRPV4 channels [132]. The authors suggested a pathway from TRPV4 channels activated by increased matrix stiffness to increased nuclear translocation of the α-SMA transcription coactivator (MRTF-A) [132]. Moreover, TRPV4 is also important for epithelial mesenchymal transition (EMT) [8], a process which may be able to differentiate more fibroblasts from epithelial cells to support fibrosis [133]. 

We were able to demonstrate an up-regulation of **TRPC6** channels in murine lung fibroblasts after the application of TGF-β1 [134], similar to cardiac fibroblasts [135], and TRPC6−/− mice were partially protected from bleomycin-induced lung fibrosis. TRPC6-deficient fibroblasts stimulated by TGF-β1 produced less collagen and showed a reduced Ca^2+^-induced translocation of nuclear factor of activated T-cells (NFAT), which regulates the transcription of profibrotic genes in mouse fibroblasts [134].

Most interestingly, **TRPA1** channels are highly expressed in primary human lung fibroblasts and are down-regulated by the application of TGF-β1 (see https://www.biorxiv.org/content/10.1101/2022.04.12.488008v2, accessed date: 14 September 2022). The Si-RNA-mediated knock-down of TRPA1 increased the expression of profibrotic genes, while the stimulation of TRPA1 by its specific activator (allylisothiocyanate (AITC)) reduced transcriptional activity, most probably by an inhibitory linker phosphorylation of SMAD 2 (see https://www.biorxiv.org/content/10.1101/2022.04.12.488008v1, accessed date: 14 September 2022). Similar results were obtained in MRC5 cells and human myofibroblasts, where TRPA1 channel activation reduced the induction of fibrotic genes or induced cell death, respectively [136,137]. Thus, TRPA1 activity seems to inhibit fibroblast-to-myofibroblast differentiation, a hallmark of lung fibrosis development in clear contrast to TRPV4 and TRPC6 channels, which support fibrotic processes.

In addition to mechanical injuries of the lung, radiation may also induce lung fibrosis. For patients with lung breast and esophageal cancer, thoracic radiotherapy is a central part of a multi-modal treatment concept [138]. However, up to 30% of patients receiving radiotherapy develop radiation-induced pneumonitis and are also at high risk of suffering from radiation-induced lung fibrosis [139]. Therefore, radiation-induced lung fibrosis (RILF) is an important dose-limiting factor with a direct impact on patient outcomes and quality of life (reviewed in [140]). **TRPM2** channels are expressed in the endothelium and in several immune cells, invading the lung after injury (see Figure 1). They harbor a NUDT9-H region, which binds ADP-ribose (ADPr) at multiple sites and is directly involved in channel gating (reviewed in [141]). Moreover, ADPr is also generated in the cell nucleus through the activation of the poly(ADP-ribose) polymerase-1 (PARP-1) by DNA damage and released from mitochondria by H_2_O_2_ [141]. Irradiation of tissues can produce both ROS-like H_2_O_2_ as well as damage to genomic DNA in the cell nucleus [142]. Most interestingly, TRPM2-deficient mice were protected from irradiation-induced salivary gland dysfunction in contrast to WT mice, which irreversibly lost salivary gland fluid secretion [143]. Both a free radical scavenger and a PARP-1 inhibitor attenuated the irradiation-induced activation of TRPM2 and induced significant recovery of salivary fluid secretion [143]. Therefore, TRPM2 channels may also serve similar functions in RILF, which needs to be analyzed in the future. 

## 5. Roles for TRP Channels in Barrier Function of the Pulmonary Endothelium

The endothelium is next to the tracheal, bronchial and alveolar epithelium, the other barrier, which protects lung tissues from bacteria, viruses and toxicants circulating in the blood. Moreover, immune cells from the blood invade lung tissues through a more permeable endothelium during infection and may cause inflammation and lung damage, which results in flooding of the alveolar space by protein-rich fluid during edema formation (reviewed in [144]). A tight regulation of lung barrier function by the endothelium is therefore crucial for the protection of the lung and has already been studied extensively over the last decades. TRPV4, TRPC6 and TRPM2 channels were identified as major players, as their Ca^2+^ influx activates the CaM (calmodulin)/MLCK (myosin light chain kinase)-signaling pathway and rearranges the cytoskeleton increasing endothelial permeability (reviewed in [4,145]). 

**TRPV4** channels are important for increasing endothelial permeability as TRPV4 activators, such as 4α-phorbol esters, initiate lung edema [146]. In these molecular processes, TRPV1 channels and protease-activated receptor 1 (PAR1) are also involved [147]. Along these lines, the TRPV4 blocker GSK2193874 was effective in inhibiting edema formation by high pulmonary venous pressure as well as in a myocardial infarction mouse model [148]. Therefore, TRPV4 channel modulators may be also successful in the treatment of pulmonary edema during the recent COVID-19 pandemic (reviewed in [149]).

As outlined above, **TRPM2** channels are expressed in the lung endothelium and participate in the regulation of barrier function [150,151], cell death, migration and angiogenesis (reviewed in [152]). Vascular endothelial growth factor (VEGF) in endothelial cells drives their migration, proliferation, and disassembly of adherens junctions, as well as the production of reactive oxygen species (ROS), which activate TRPM2 channels [152]. Importantly, the deletion of endothelial TRPM2 channels reduce the transendothelial migration of polymorphonuclear neutrophils, which produce ROS in response to LPS from Gram-negative bacterial membranes in the circulating blood of the pulmonary vasculature [153].

LPS also activates toll-like receptors (e.g., TLR4) in endothelial cells, and **TRPC6** channels are important for linking LPS via TLR4 to endothelial permeability [154]. Therefore, TRPC6 and TRPM2 channel inhibition are important therapeutic options to reduce vascular barrier disruption and inflammation due to endotoxins. 

Major endogenously produced molecules regulating endothelial permeability are platelet activating factor (PAF) and prostaglandins (e.g., PGE_2_) (reviewed in [4]). PAF is a critical mediator in acute lung injury, and its concentration is elevated in the lungs of patients with acute respiratory distress syndrome (ARDS) [155]. An involvement of TRPC6 channels in the complex PAF/PGE_2_ signaling pathways was recently dissected on a molecular basis [156,157]. PAF binding to its receptor activates acid sphingomyelinase, which produces ceramide. The formation of ceramide increases caveolin 1, endothelial NO-synthase (eNOS) and TRPC6 channels in microdomains called caveolae [156]. The inactivation of eNOS by ceramide results in reduced NO production and decreased cGMP-mediated phosphorylation of a threonine residue in the TRPC6 amino-terminus, which inhibits channel activity [156]. The subsequent activation of TRPC6 channels results in increased endothelial permeability and vascular leakage [156]. Most interestingly, the PAF response was decreased in mouse models deficient for the prostaglandin-receptor 3 (EP3R), and EP3R-ligands, such as sulprostone, were able to increase PAF-mediated endothelial permeability in WT but not in TRPC6-deficient mice [157]. Indeed, ligand coupling to EP3 receptors was also able to stimulate TRPC6 activity by Src-kinase-mediated phosphorylation of a tyrosine residue [157]. 

During lung transplantation, I/R-induced injury is a major reason for graft failure [158]. Next to toxicant-induced edema [159], I/R-induced edema can be mimicked in the isolated perfused and ventilated mouse lung (IPVML) [160]. Most interestingly, I/R-induced edema was absent in TRPC6-deficient mice, and edema formation in double TRPC6/TRPV4−/− lungs was similar to WT mice [119], which may be explained by an antagonistic function of TRPC6 and TRPV4 in the lung endothelium and epithelium, respectively (reviewed in [8]). Most interestingly, selective TRPC6 antagonists, such as larixyl *N*-methylcarbamate, reduces I/R-induced edema formation in IPVML [161] and may also protect human lungs intended for transplantation in the future. A recent clinical study by the pharmaceutical company Boehringer-Ingelheim with another TRPC6 antagonists (BI 764198) in prevention/progression of ARDS and ARDS-related complications secondary to COVID-19 (see https://clinicaltrials.gov under NCT04604184, accessed date: 14 September 2022) was, however, interrupted in 2022 due to lack of efficiency (https://www.boehringer-ingelheim.de/covid-19/kampf-gegen-covid-19/forschungsupdate, accessed date: 14 September 2022). 

## 6. TRP Channels and the Development of Pulmonary Arterial Hypertension

In addition to regulating endothelial barrier permeability, certain TRP channels have also been implicated in endothelial dysfunction, leading to pulmonary arterial hypertension (PAH). Characterized by vasoconstriction, increased pulmonary arterial pressure, oxidative stress, inflammation, and severe remodeling of the pulmonary vasculature, PAH can lead to right heart failure and premature death [162]. It is established that endothelial Ca^2+^ signaling is essential in vascular remodeling. TRP channels involved in angiogenesis and the formation of arteries and the vasculature are also crucial for cell proliferation and migration in the pathogenesis of PAH [163].

An important role for TRPC6 channels in the development of PAH has been extensively investigated in the last 18 years. While we identified no discernable differences in chronic (3 weeks) hypoxia-induced PAH (and the accompanying pulmonary vascular remodeling) in TRPC6-deficient and wild-type mice, other scientists reported that TRPC6-deficient mice exposed less PAH and pulmonary vascular remodeling after 4 weeks of hypoxia (reviewed in [164]). Interestingly, a very recent study showed that administration of a TRPC6-specific antagonist reversed existing PAH in mice by nearly 50%, as evidenced by the regression of pulmonary vascular remodeling [165]. 

Although most of the findings focus on TRPC6, other TRPC members might also be important for the development of PAH. TRPC1 has also been linked to excessive proliferation of pulmonary arterial smooth muscle cells (PASMC), and may be directly responsible for the elevated Ca^2+^ currents in proliferating PASMC [166] and reviewed in [164]. In addition to TRPC6, PASMC from patients with idiopathic PAH showed also significantly elevated mRNA and protein expression of TRPC3 channels when compared to those from normotensive patients [167]. The role of TRPC4 in PAH was investigated in a model of hypoxic PAH in which rats were pretreated with a vascular endothelial growth factor receptor 2 inhibitor, exposed to three weeks of hypoxia, and then returned to normoxia. In this model, TRPC4 contributed to the hyper-permeability of endothelial cells, demonstrating a novel type of endothelial abnormality in severe experimental PAH [168]. 

Moreover, recent research has connected ROS-induced activation of TRPM2 to the enhanced proliferation and migration of PASMC in a model of chronic hypoxic pulmonary hypertension [169]. Increased concentrations of intracellular ROS also triggered the opening of TRPV4 in microvascular endothelial cells inducing endothelial dysfunction and the development of PAH in a rat model [170]. 

In a recent review, the role of TRPV1 as a modulator of pro- and anti-inflammatory neuropeptides in the development of PAH was summarized [171], and TRPV1 activation has also been shown to induce changes in cellular proliferation rates upon ROS exposure in endothelial precursor cells after photo-stimulation [172]. Therefore, multiple TRP channels may be involved in the different stages of PAH.

## 7. Conclusions

In this review, we summarized recent data on TRP channel function in non-neuronal tissues of the respiratory system. In most of the cells, TRP channels are expressed, and strong evidence for an essential role in airway function and associated diseases, such as CF, asthma, fibrosis and edema was demonstrated for TRPA1, TRPC6, TRPM2, TRPM5, TRPM7, TRPV2, TRPV4 and TRPV6. Despite these promising data, clinical trials with TRP modulators were so far not successful, except for capsaicin patches and injections acting on the neuronal expressed TRPV1 channels ([173], reviewed in [174]. Therefore, groundbreaking research on this channel was rewarded with the Nobel prize in 2021 [175]. By continuing efforts to understand TRP channel function, drugs acting on other TRP channels in the respiratory system will hopefully be invented in the near future. 

## Figures and Tables

**Figure 1 cells-11-02907-f001:**
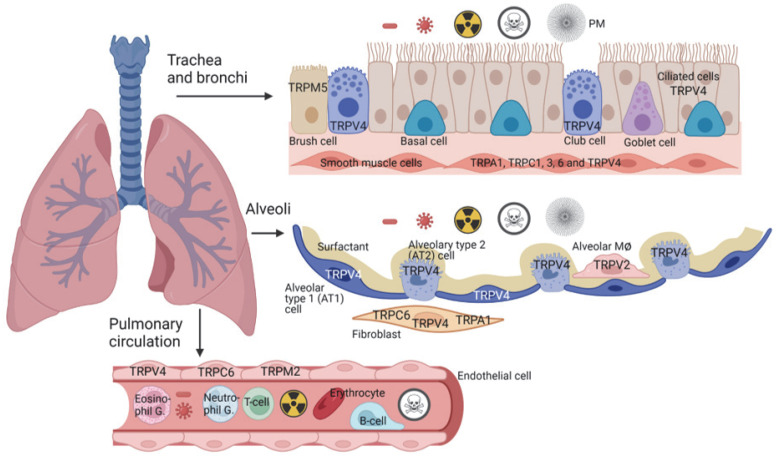
Cells of the human and murine respiratory tract and TRP channel function. The upper part of the lower airways with basal, goblet, club and ciliated cells as well as the alveolus and the pulmonary vasculature are shown. Functional active TRP channels based on reported data are indicated. In the very specialized brush cells TRPM5 channels are exclusively active. In the alveoli, alveolar type 1 (AT1) cells are important for barrier function, while AT2 cells secrete a surfactant that can renew damaged AT1 cells. Alveolar and interstitial macrophages (MØ) are able to phagocytize microorganisms and particulate matter (PM). Fibroblasts involved in wound healing own TRPC6 and TRPA1 proteins. The latter channel, however, exists only in human and not in mouse fibroblasts. Endothelial cells as natural barrier to exclude toxicants perfused in the blood need active TRPC6, TRPM2 and TRPV4 channels. PM, particulate matter. See also Table 1 for single-cell mRNA expression data and the text for more details.

**Table 1 cells-11-02907-t001:** Single-cell mRNA expression levels of TRP channels in the respiratory tract.

Cell Type	TRP Channel	nTPM
AT2 cells	TRPC6	2.6
	TRPM2	0.5
B cells	TRPM2	2.0
	TRPV2	**11.1**
	TRPV4	0.7
Basal cells	TRPV4	**17.1**
Ciliated cells	TRPC6	0.8
	TRPV2	0.8
	TRPV4	**28.2**
Club Cells	TRPC6	**10.5**
	TRPV2	0.6
	TRPV4	**12.0**
Endothelial cells	TRPC6	1.6
	TRPM2	2.7
	TRPV2	**12.3**
	TRPV4	1.4
Fibroblasts	TRPA1	8.1
	TRPC6	3.1
	TRPM2	0.5
	TRPV2	**11.0**
	TRPV4	1.2
Macrophages	TRPC6	1.8
	TRPM2	**16.7**
	TRPV2	**28.7**
	TRPV4	2.6
Smooth muscle cells	TRPA1	0.8
	TRPC6	**20.2**
	TRPM2	0.4
	TRPV2	**23.6**
	TRPV4	1.1
T cells	TRPC6	0.1
	TRPM2	4.2
	TRPV2	**26.8**
	TRPV4	0.3

nTPM (**high** =>10), normalized transcripts per million from the Human Protein Atlas. (www.proteinatlas.org, accessed date: 9 September 2022).

## Data Availability

Not applicable.

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
