# Peer review of "Transient Receptor Potential (TRP) Channels in Airway Toxicity and Disease: An Update"

_cells, 2022, doi:10.3390/cells11182907_

Round 1

Reviewer 1 Report (Previous Reviewer 2)

Although the format of the article has not been revised to highlight recent advances in the field, the references have been updated to take into account the latest published articles.

Author Response

Reviewer 1:

Although the format of the article has not been revised to highlight recent advances in the field, the references have been updated to take into account the latest published articles.

Reply: We are confident that we now present a review which highlights recent advances in the field with the help of all reviewers.

Reviewer 2 Report (New Reviewer)

General Comments:

This manuscript reviews the current knowledge on the role of TRPs in non-neuronal airway cells. This topic is mainly described in the pathological context of asthma, COPD, cystic fibrosis and pulmonary fibrosis, and lung edema. This work also focuses on the authors' recent findings on TRPs. Nevertheless, the involvement of TRPs in pulmonary hypertension and lung cancer, two diseases related to disruption of endothelial barrier function and immune regulation, is not addressed. Overall, the manuscript is well written and provides a wealth of mechanistic information; however, some assertions and interesting findings could be further elaborated to present a more complete revision. The material presented herein is topical, and much of the work described and discussed is very recent, so a review of the subject is warranted and should be of great interest to readers of this journal. Specific comments are included below.

Specific Comments:

Several paragraphs are marked in yellow along the manuscript, and I do not understand the meaning of this action.

1. Lines 34-35. Na+ permeability is also involved in TRP channel function. Therefore, I suggest adding this line: "TRPs as functional channels are important for cellular Ca2+ and Na+ homeostasis".

2. Line 39. Authors should include the isoforms in each family and define the abbreviations.

3. Lines 46-50. The aim of the review is to provide an up-to-date overview of the function of TRPs in respiratory diseases in the context of immune regulation and barrier function, but two well-documented illnesses are missing. For example, TRPC proteins have been suggested to contribute to the development of pulmonary hypertension because they are expressed in pulmonary artery smooth muscle cells and endothelial cells and contribute to Ca2+ influx, smooth muscle contraction, proliferation, and remodeling (see for example: doi.org/10.3389/fphys.2019.01618; doi.org/10.1177/2045894018798569; doi.org/10.3389/fimmu.2017.00707; doi.org/10.1073/pnas.0405908101; doi.org/10.1164/rccm.201307-1252OC). In addition, TRP proteins have been shown to play a role in lung cancer by participating in cell defense mechanisms and influencing cell survival after exposure to toxic compounds by controlling apoptotic signals (see for example: doi.org/10.3390/ph11040090; doi.org/10.3389/fonc.2022.773654; doi.org/10.1002/jcla.24229; doi.org/10.3390/medsci7120108).

4. Lines 96-107. Authors argue that compounds like bitter tasting denatonium, PQS, acyl homoserine lactones, and formylated bacterial peptides stimulate PLCβ2 to produce IP3, which releases Ca2+. This Ca2+ increment will activate TRPM5 with the subsequent Na+ influx. However, ciliated cells in TRPM5-/- mice showed no increases in [Ca2+]i and reduced ciliary motility after the application of bacterial triggers. Authors should explain this phenomenon better since bacterial triggers acting by unknown receptors will increase by themself the [Ca2+]I. 

5. Line 104. Delete ACh after acetylcholine.

6. In the section on TRPs and cystic fibrosis, the work by Prandini and colleagues should be considered (doi.org/10.1165/rcmb.2016-0089OC, PubMed: 27281024).

7. Line 188. Insert a comma between IL-13 and eotaxin : "IL-13, eotaxin…"

8. Line 299. Particle size should be < 5 µm. In reference 8 of this review, Dietrich et al, Cell calcium, 2017, page 179: 4. Toxic inhalation hazards…., second paragraph, authors wrote that particles >5 µm in diameter are trapped in the mucus coating the epithelium of the upper airways; thus they do not reach deep lung compartments.

9. Line 336. Delete calcium and add Ca2+

10. Line 405. Add idiopathic pulmonary fibrosis (IPF).

11. Line 414. Delete idiopathic pulmonary fibrosis.

12. Lines 440-457. The radiation-induced increase in the production of ROS has been associated with the activity of TRPs (see, for example: doi.org/10.1155/2020/8871745; doi.org/10.1007/s10787-021-00802-1). In addition, ROS has been frequently associated with lung cancer (see, for example: doi.org/10.1016/j.bbamcr.2021.118950), a good reason to include about TRP proteins and lung cancer.

13. Lines 479-486. It is well known that abnormal formation of ROS in endothelial and vascular smooth muscle can lead to cell proliferation and vascular remodeling (as indicated by the authors in lines 482-483). Furthermore, these changes in vascular physiology lead to pulmonary hypertension. The role of TRP channels in the development of pulmonary hypertension has already been studied (see for example: DOI: 10.1016/j.vph.2021.106860), so a section on this topic should be considered.

Minor Comments:

 Line 199. Delete inhibitor and add antagonist as in line 198.

 Line 285. Delete inhibitors and add antagonists.

 Line 290. Delete inhibition and add antagonism.

 Line 292. Delete inhibitor and add antagonist.

 Line 294. Delete inhibitors and add antagonists.

 Line 514. Delete inhibitors and add antagonists.

 Line 517. Delete inhibitor and add antagonist.

Author Response

Reviewer 2:

Several paragraphs are marked in yellow along the manuscript, and I do not understand the meaning of this action.

Reply: The yellow color was added by the editorial office and not by the authors.

  1. Lines 34-35. Na+ permeability is also involved in TRP channel function. Therefore, I suggest adding this line: "TRPs as functional channels are important for cellular Ca2+ and Na+ homeostasis".

Reply: We altered the sentence as suggested by adding Na+ in line 36 of the revised manuscript. 

  1. Line 39. Authors should include the isoforms in each family and define the abbreviations.

Reply: We defined abbreviations and included the isoforms in each family (line 40-43).

  1. Lines 46-50. The aim of the review is to provide an up-to-date overview of the function of TRPs in respiratory diseases in the context of immune regulation and barrier function, but two well-documented illnesses are missing. For example, TRPC proteins have been suggested to contribute to the development of pulmonary hypertension because they are expressed in pulmonary artery smooth muscle cells and endothelial cells and contribute to Ca2+ influx, smooth muscle contraction, proliferation, and remodeling (see for example: doi.org/10.3389/fphys.2019.01618; doi.org/10.1177/2045894018798569; doi.org/10.3389/fimmu.2017.00707; doi.org/10.1073/pnas.0405908101; doi.org/10.1164/rccm.201307-1252OC). In addition, TRP proteins have been shown to play a role in lung cancer by participating in cell defense mechanisms and influencing cell survival after exposure to toxic compounds by controlling apoptotic signals (see for example: doi.org/10.3390/ph11040090; doi.org/10.3389/fonc.2022.773654; doi.org/10.1002/jcla.24229; doi.org/10.3390/medsci7120108).

Reply: The reviewer correctly mentioned the importance of discussing the relationship between TRP channels and the pathogenesis of pulmonary hypertension and lung cancer. In order to provide now a well-rounded review of the function of TRP channels in lung disease, we have added a section describing the current state of research on TRP channels, its activation by hypoxia/ROS (line 358-372), and the vascular remodeling leading to pulmonary hypertension. We also added a section on TRP channel function in the development, progression and therapy of lung cancer (lines 372-389 of the revised manuscript).

  1. Lines 96-107. Authors argue that compounds like bitter tasting denatonium, PQS, acyl homoserine lactones, and formylated bacterial peptides stimulate PLCβ2 to produce IP3, which releases Ca2+. This Ca2+ increment will activate TRPM5 with the subsequent Na+ influx. However, ciliated cells in TRPM5-/- mice showed no increases in [Ca2+]i and reduced ciliary motility after the application of bacterial triggers. Authors should explain this phenomenon better since bacterial triggers acting by unknown receptors will increase by themself the [Ca2+]I.

Reply: This sentence is indeed confusing for the reader. To avoid an extensive discussion on a rise in [Ca2+]i induced by bacterial triggers, which would unnecessarily further extend this review, we deleted “showed no increases in [Ca2+]i.” (line 110-111).

  1. Line 104. Delete ACh after acetylcholine.

Reply: We deleted ACh in line 108 of the revised manuscript.  

  1. In the section on TRPs and cystic fibrosis, the work by Prandini and colleagues should be considered (doi.org/10.1165/rcmb.2016-0089OC, PubMed: 27281024).

Reply: The work of Prandini and colleagues is indeed an excellent addition for an involvement of TRP channels in cystic fibrosis. We thank the reviewer for this suggestion and added a paragraph describing this work in lines 147-155 of the revised manuscript.

  1. Line 188. Insert a comma between IL-13 and eotaxin : "IL-13, eotaxin…"

Reply: We inserted a comma (line 200 of the revised manuscript).

  1. Line 299. Particle size should be < 5 µm. In reference 8 of this review, Dietrich et al, Cell calcium, 2017, page 179: 4. Toxic inhalation hazards…., second paragraph, authors wrote that particles >5 µm in diameter are trapped in the mucus coating the epithelium of the upper airways; thus they do not reach deep lung compartments.

Reply: We apologize for this mistake and changed it into <5 µm (line 311 of the revised manuscript).

  1. Line 336. Delete calcium and add Ca2+.

Reply: We deleted calcium and added Ca2+ (line 348 of the revised manuscript).

  1. Line 405. Add idiopathic pulmonary fibrosis (IPF).
  2. Line 414. Delete idiopathic pulmonary fibrosis.

Reply: We defined “idiopathic pulmonary fibrosis” in line 455 and used the abbreviation in line 464 of the revised manuscript.

  1. Lines 440-457. The radiation-induced increase in the production of ROS has been associated with the activity of TRPs (see, for example: doi.org/10.1155/2020/8871745; doi.org/10.1007/s10787-021-00802-1). In addition, ROS has been frequently associated with lung cancer (see, for example: doi.org/10.1016/j.bbamcr.2021.118950), a good reason to include about TRP proteins and lung cancer.

Reply: We cited these important manuscripts in lines 371-372 and included a paragraph on lung cancer in lines 379-389 of the revised manuscript.

  1. Lines 479-486. It is well known that abnormal formation of ROS in endothelial and vascular smooth muscle can lead to cell proliferation and vascular remodeling (as indicated by the authors in lines 482-483). Furthermore, these changes in vascular physiology lead to pulmonary hypertension. The role of TRP channels in the development of pulmonary hypertension has already been studied (see for example: DOI: 10.1016/j.vph.2021.106860), so a section on this topic should be considered.

Reply: We added a new paragraph on the involvement of TRP channels in pulmonary arterial hypertension (lines 574-611).

Minor Comments:

 Line 199. Delete inhibitor and add antagonist as in line 198.

 Line 285. Delete inhibitors and add antagonists.

 Line 290. Delete inhibition and add antagonism.

 Line 292. Delete inhibitor and add antagonist.

 Line 294. Delete inhibitors and add antagonists.

 Line 514. Delete inhibitors and add antagonists.

 Line 517. Delete inhibitor and add antagonist.

Reply: These changes were included in lines 211, 297, 299, 304, 306, 565 and 568 of the revised manuscript.

Reviewer 3 Report (New Reviewer)

This is a relatively comprehensive review of the potential role of TRP channels in airway disease. My major comment relates to the exclusion of TRPML3 which was the subject of a recent high impact paper (Spix et al, Nat Comm 2022) implicating the channel in emphysema; this should either be included or minimally the recent review in this journal be cited (Spix et al, Cells 2022, 11, 304).

Minor comments:

Line 30: immune cells invading lung  tissue can result in a variety of respiratory pathologies not just ARDS; this sentence needs to be changed to reflect this please.

Line 91: please use either brush of tuft for simplicity after highlighting the alternate nomenclature.

Author Response

Reviewer 3:

My major comment relates to the exclusion of TRPML3 which was the subject of a recent high impact paper (Spix et al, Nat Comm 2022) implicating the channel in emphysema; this should either be included or minimally the recent review in this journal be cited (Spix et al, Cells 2022, 11, 304).

Reply: In our previous version, we had not included any lysosomal channels and therefore, did not include the highly relevant publication from Spix et al. in 2022. We now cite this publication and refer the readers to the respective review for further information on lysosomal TRP channels in airway diseases. Lines 430-435.

Minor comments:

Line 30: immune cells invading lung tissue can result in a variety of respiratory pathologies not just ARDS; this sentence needs to be changed to reflect this please.

Reply: We added “amongst other diseases” to reflect that ARDS is not the only respiratory pathology resulting from invading immune cells in line 31 of the revised manuscript. 

Line 91: please use either brush of tuft for simplicity after highlighting the alternate nomenclature.

Reply: We deleted “tuft” (line 96, 97) after highlighting the alternate nomenclature (line 95 of the revised manuscript). 

Round 2

Reviewer 2 Report (New Reviewer)

The authors' reply substantially improves the review; therefore, I do not have additional comments. 

This manuscript is a resubmission of an earlier submission. The following is a list of the peer review reports and author responses from that submission.

Round 1

Reviewer 1 Report

The authors present a nice and readable update on TRP channels regarding airway toxicity and disease. I believe it is suitable for publication as it is, however, and for the sake of context, I would recommend possible modifications that hopefully are perceived as improvements.

First, I think it is wise to indicate in Figure 1, or as another Figure or a table, the levels of expression (RNA and/or protein) of each channel in the specific cell/tissue type (refer to the protein atlas, or GEO databases) in order to get a clearer picture of the relevance of each channel. This line of thought goes with the fact that especially in the field of TRP channels, the knowledge on protein function is not related to the relevance of the channel in the specific function. For instance, the review has plenty of information on TRPV4, because there is a huge amount of information derived from a wide pharmacological toolbox for this channel, as compared to other TRP channels. I think expression levels would create a context for the overview of all TRP channels in airway toxicity and disease, regardless there is available/published information on that channel. 

Another aspect that could be benefitial for readers in not so favoured areas is to use open access or accessible reviews rather than book chapters (when availabel); as a reviewer I could not get open access to 104, I believe there are extensive reviews on TRPV2 and other channels in open access sources.

Reviewer 2 Report

This review summarizes the expression and functions of selected transient receptor potential (TRP) channels (TRPA1, TRPC6, TRPM2, TRPM5, TRPV2 and TRPV4) in the airways, and their role in lung toxicity and diseases such as asthma or cystic fibrosis.

Although this paper is well written, it is difficult to draw a clear message about the role of the different TRPs in the airways. Maybe too many TRPs and toxic responses/diseases are covered at once, for the reader to appreciate the scope of this review.

Besides, the authors have published 4 reviews on the same topic since 2017. They present the actual article as "an update", but they do not highlight which knowledge is new since their last review in 2021. Is there enough publications in the field, in a so short time, to justify an update? Most cited articles are not so recent, whereas some recent articles in the field are not cited such as :

- The Transient Receptor Potential Channel Vanilloid 1 Is Critical in Innate Airway Epithelial Responses to Protease Allergens (Schiffers et al, 2020)

- Transient receptor potential cation channel subfamily V (TRPV) and its importance in asthma (Reyes-García et al., 2022)

- The transient receptor potential vanilloid 4 (TRPV4) ion channel mediates protease activated receptor 1 (PAR1)-induced vascular hyperpermeability (Peng et al., 2020);

- Roles of TRPA1 and TRPV1 in cigarette smoke-induced airway epithelial cell injury model (Wang et al., 2019)

- TRPV1 and TRPA1 in Lung Inflammation and Airway Hyperresponsiveness Induced by Fine Particulate Matter (PM2.5) (Xu, et al. 2019).